# ODEStream: A Buffer-Free Online Learning Framework with ODE-based Adaptor for Streaming Time Series Forecasting

**Futoon M.Abushaqra**                                      *futoon.abushaqra@rmit.edu.au*
*School of Computing Technologies*
*RMIT University*

**Hao Xue**                                                 *hao.xue1@unsw.edu.au*
*School of Computer Science and Engineering*
*University of New South Wales*

**Yongli Ren**                                              *yongli.ren@rmit.edu.au*
*School of Computing Technologies*
*RMIT University*

**Flora D.Salim**                                           *flora.salim@unsw.edu.au*
*School of Computer Science and Engineering*
*University of New South Wales*

**Reviewed on OpenReview:** *https://openreview.net/forum?id=TWOTKhwU5n*

## Abstract

Addressing the challenges of irregularity and concept drift in streaming time series is crucial for real-world predictive modelling. Previous studies in time series continual learning often propose models that require buffering long sequences, potentially restricting the responsiveness of the inference system. Moreover, these models are typically designed for regularly sampled data, an unrealistic assumption in real-world scenarios. This paper introduces ODEStream, a novel buffer-free continual learning framework that incorporates a temporal isolation layer to capture temporal dependencies within the data. Simultaneously, it leverages the capability of neural ordinary differential equations to process irregular sequences and generate a continuous data representation, enabling seamless adaptation to changing dynamics in a data streaming scenario. Our approach focuses on learning how the dynamics and distribution of historical data change over time, facilitating direct processing of streaming sequences. Evaluations on benchmark real-world datasets demonstrate that ODEStream outperforms the state-of-the-art online learning and streaming analysis baseline models, providing accurate predictions over extended periods while minimising performance degradation over time by learning how the sequence dynamics change. The implementation of ODEStream is available at: `https://github.com/FtoonAbushaqra/ODEStream.git`.

## 1 Introduction

In today's data-driven world, the rapid growth of interconnected devices and systems has led to an explosion of time series data generated in real time. These data streams offer invaluable insights across diverse domains, including finance (Kovacs et al., 2021; Sezer et al., 2020), healthcare (Zhang1a et al., 2021; Kaushik et al., 2020), environmental monitoring (Depuru et al., 2011), and industrial processes (Maschler et al., 2020). Being able to analyse the streaming data in real-time provides more accurate and up-to-date information, allowing us to react to changes as they happen and adding a significant value to the outputs (Almeida et al., 2023).

Handling streaming analysis requires an effective continual online learning approach. Recently few studies have addressed streaming analysis for regression and forecasting (Cossu et al., 2021), by relying on buffer memory to apply a replay method for continual learning (Matteoni et al., 2022; Kwon et al., 2021; Kiyasseh et al., 2021; Xiao et al., 2022; Chen et al., 2021). Using the replay method, a training set initialises the model's parameters, and a selected subset of this data, as well as future incoming data samples, is saved to be used later when the model is retrained (Hoi et al., 2021; Fekri et al., 2021; He, 2021). This process focuses on avoiding catastrophic forgetting issues by continuously updates the model knowledge while retraining on both incoming and historical data. However, since this method is effective at preserving previously learned information, its performance on time series data is uncertain. By focusing mainly on avoiding forgetting, it may fail to adequately capture new and evolving patterns in the data over time (Cossu et al., 2021; Pham et al., 2023). Additionally, these replay-based frameworks introduce significant complexity, as they must carefully determine which data samples to store in the buffer, when to update importance weights, and how to allocate memory and computational resources efficiently. These challenges highlight the need for a more streamlined and efficient approach to managing streaming time series data. Furthermore, the impact of catastrophic forgetting differs significantly in time series data and predictive modelling compared to non-temporal data and classification tasks, where class- and task-incremental learning are more common scenarios (Ao & Fayek, 2023; Pham et al., 2023). In such scenarios, the model's objective is to learn new tasks or classes while ensuring that previously learned ones are not forgotten. On the other hand, when developing a lifelong learning model for streaming time series data, a key consideration emerges: to what extent should we prioritize preserving historical information, especially when that data may have evolved or changed over time (Cossu et al., 2021; Pham et al., 2023).

Streaming time series forecasting faces significant challenges, including **(1)** ***Concept Drift Phenomenon***, where underlying patterns or relationships within the data change over time, posing a substantial challenge for predictive modelling (Fekri et al., 2021). This issue extends beyond online learning models to impact batch learning approaches, as models initially proven accurate can quickly become obsolete as the data continues to evolve. **(2)** ***Temporal Irregularity***, which is common in streaming data, especially in real-world applications where the data lacks a specific time frame. **(3)** ***Balanced Adaptation***, where maintaining the right balance between historical and recent data is important to preventing biases towards outdated data while effectively adapting to the dynamic nature of streaming time series data. Therefore building a robust framework that can adeptly address these challenges is essential for maintaining model accuracy over time.

This paper aims to overcome the restrictive boundaries of streaming time series processing by introducing a novel online learning model (ODEStream). Our focus lies on enabling the model to adapt to evolving data patterns without relying on a complex framework while maintaining good performance over time. Specifically, we investigate how the novel neural ordinary differential equation (ODE) methods can be leveraged for memory-free online forecasting for streaming time series data. The key contributions of this paper are as follows:

- **Unique Perspective:** We redefine the context of continuous time series forecasting problem by introducing a novel perspective that prioritises learning the evolving dynamics and data distribution, rather than relying on the preservation of historical samples. This shift enables our model to adapt more effectively to changes over time, offering improved performance compared to traditional approaches.

- **Utilising Neural ODEs:** To the best of our knowledge, this work is the first to leverage neural Ordinary Differential Equation (ODE) models for continuous learning and online forecasting. Due to their ability to dynamically adapt and provide a continuous hidden state, ODEs offer a more flexible approach compared to traditional discrete-time models, especially in handling irregularly sampled data.

- **Streamlined Buffer-Free Framework:** We introduce ODEStream, a buffer-free continual learning framework designed to have a straightforward and efficient training process. Unlike existing frameworks that require complex decision-making for upcoming data; such as defining thresholds, trigger values, or managing buffered samples, ODEStream eliminates these requirements entirely.

This streamlined design reduces overhead, simplifies training, and improves the model's usability in real-world streaming scenarios.

- **Comprehensive Evaluation:** We evaluate ODEStream against state-of-the-art continual learning models and demonstrate that it effectively mitigates the impact of irregularities, adapts to data drift, and maintains stable performance over extended streaming periods.

## 2 Related Work and Background

### 2.1 Continual Learning

Continual learning, also known as incremental learning or lifelong learning in several articles (Ao & Fayek, 2023), is the process of training a model on a sequence of tasks over time without forgetting previously learned tasks (Lopez-Paz & Ranzato, 2017). The method should maintain a balance between preserving the knowledge acquired from prior tasks and acquiring new knowledge for future tasks while having restricted access to historical experiences (Grossberg, 2013). There are several scenarios for continual learning systems, including Instance-Incremental Learning (IIL), Task-Incremental Learning (TIL), and Online Continual Learning (OCL), among others (Wang et al., 2023). Regardless of the scenario in which continual learning is applied, the strategies and methods that have been used are summarised as Regularization, Replay, Optimisation, and Representation-based approaches (Wang et al., 2023).

Replay-based, also known as Memory-based approaches, is the most common for various fields and has proven to be highly effective. In these approaches, a subset of past examples is saved and can be replayed when the model is trained in the future, relying on dynamic external memory. An example of this approach is Experience Replay (ER), which relies on a fixed-sized replay buffer (Lopez-Paz & Ranzato, 2017; Chaudhry et al., 2019a). ER methods are commonly used in reinforcement learning (Lin, 1992; Rolnick et al., 2019; Foerster et al., 2017), but they have also been applied to supervised learning (Chaudhry et al., 2019b; Riemer et al., 2019; Buzzega et al., 2020).

Regularization approaches (Li & Hoiem, 2017; Rebuffi et al., 2017; Kirkpatrick et al., 2017; Aljundi et al., 2018) add explicit regularization terms to balance the learning of the old and new tasks. They work by regulating the learning process through either penalising feature drift on previously learned tasks or reducing changes in previously learned parameters (Li & Hoiem, 2017; Rebuffi et al., 2017; Kirkpatrick et al., 2017; Aljundi et al., 2018). Regularization methods often require storing a static version of the old model for reference (Wang et al., 2023). Finally, Model-based methods are applied by changing network structure or applying multiple models to respond to different tasks (Fernando et al., 2017; Mallya & Lazebnik, 2018). This method faces a challenge with the potential growth of network parameters. A comprehensive survey on general continual learning methods is provided in (Wang et al., 2023).

While extensive work has been conducted on continual learning, there has been limited focus on its application to time series. Recently, In (Sun et al., 2022), a novel concept known as Continuous Classification of Time Series (CCTS) was introduced, driven by the necessity for early classification of time series, as highlighted by Gupta et al. (2020). Following this, a recent article has addressed the issues of CCTS (Sun et al., 2023). To evaluate the continuous learning methods on time series data, Maschler et al. (2021; 2022) put several regularization methods undergoing testing for time series anomaly detection and classification; both studies' finding indicates that Online Elastic Weight (OEW) outperformed other regularization methods. In (Matteoni et al., 2022), the authors proposed new benchmark datasets for time series continual learning for classification on human state monitoring; using the proposed data, the authors tested several continual learning methods and found that experience replay proves to be the most efficient approach, A similar conclusion was reached in (Kwon et al., 2021), where the main goal of the study is to check if methods developed for image-based continual learning classification will work on time series. The authors explored different replay and regularization-based methods on six datasets. The study found that if storage is not an issue, then the replay method gives the best performance. Another comparison of continual learning methods was applied in (Cossu et al., 2021) where RNN's performance for continual learning was explored across various scenarios.

In (Maschler et al., 2020), the authors presented a model for continuous learning fault prediction. The method was applied to the NASA turbofan engine dataset, and the results demonstrated the model's effective performance with distributed datasets without the need for centralised data storage. At the same time, two replay methods have recently been presented for time series (Kiyasseh et al., 2021; Xiao et al., 2022). In (Kiyasseh et al., 2021), CLOPS was proposed for clinical temporal data, (Xiao et al., 2022) proposed method for streaming traffic flow sensor data. Finally, the work (He & Sick, 2021) was the first to describe continual learning and catastrophic forgetting for regression tasks. The model utilised a neural network with buffers and was deployed with OEW consolidation. Then, the same authors extended their work and focused on power forecasting. In (Gupta et al., 2021), the author focused, for the first time, on multivariate time series and multi-task continual learning. They addressed the issues using a memory-based method for the seen data and proposed an RNN-GNN model.

## 2.2 Time Series Streaming Data

In general, online time series forecasting has primarily been studied using machine learning models (Jiménez-Herrera et al., 2023; Li et al., 2019; Chen et al., 2020; Pham et al., 2023; Melgar-García et al., 2023; Cossu et al., 2021; Fekri et al., 2021; Ao & Fayek, 2023) and statistical forecasting techniques (Alberg & Last, 2018), either using online-offline learning (Chen et al., 2020; Pham et al., 2023; Melgar-García et al., 2023), or by applying incremental learning (Alberg & Last, 2018; Li et al., 2019). However, using machine learning is not sufficient for multivariate real-world data, where the temporal dependence and pattern are complex and hard to discover. Recent works have focused on online time series forecasting using deep learning. The work in (Wambura et al., 2020) introduced a model named OFAT, which stands for One Sketch Fits All-Time Series. The method was designed for long-range forecasting in feature-evolving data streams, which involve data that changes over time and has varying features. The model employs deep neural networks to learn from a small sample of data streams and subsequently applies the learned patterns to new data streams. For that work, CNN is used to extract features from the input data, while the LSTM is used to predict future values of the output data.

In another work (Pham et al., 2023), the authors introduced a novel framework for online time series forecasting that combines the strengths of deep neural networks and Complementary Learning Systems (CLS). The framework, named FSNet, consists of two components: a slowly-learned backbone and a fast adapter. The backbone is responsible for learning long-term temporal dependencies, while the adapter is responsible for learning short-term changes. The two components are dynamically balanced to ensure that the model can adapt to both new and recurring patterns. When the data distribution changes, the adapter is updated to learn the new patterns. When the data distribution is stable, the backbone is updated to improve the accuracy of the prediction, while memory is used to store the historical data that the model has learned. Nevertheless, these works continue to face challenges when dealing with irregular samples.

More Recent studies have focused on the temporal distribution shift problem. In (Du et al., 2021), the concept of Temporal Covariate Shift (TCS) was introduced, and the AdaRNN framework was proposed. AdaRNN combines Temporal Distribution Characterisation (TDC) and Temporal Distribution Matching (TDM) to adaptively manage distribution changes. While the experimental results show that AdaRNN significantly improves forecasting accuracy, it relies heavily on correctly segmenting the data into diverse periods. Bai et al. (2022) introduced the DRAIN framework, which also focuses on adapting to changing data distributions over time by addressing the challenge of temporal domain generalization. DRAIN uses a Bayesian approach to jointly model data and neural network dynamics. It employs a recurrent graph generation scenario to encode and decode dynamic neural networks, capturing temporal drifts in both data and model parameters. A major drawback of the model is its requirement to encode and decode the entire network parameters, which adds significant complexity to the model, making it computationally intensive.

Another online learning algorithm for sequence data is Real Time Recurrent Learning (RTRL), which was recently revisited by (Irie et al., 2023) to improve its computational and memory requirements. The authors highlight RTRL's conceptual advantages over Backpropagation Through Time (BPTT) along with the time and space complexity challenges. The paper explores approximation theories and tests RTRL in realistic settings and proposes a Real-Time Recurrent Actor-Critic (R2AC) framework that uses RTRL instead of standard Backpropagation Through Time (BPTT) for RNNs.

Recent work has also focused on online variational methods for state estimation in dynamic systems (Dowling et al., 2023; Campbell et al., 2021). In (Dowling et al., 2023), the authors introduce the exponential family variational Kalman filter (eVKF), an online recursive Bayesian method that infers latent neural trajectories while learning the underlying dynamical system. The eVKF effectively handles arbitrary likelihoods and offers a closed-form variational analogue to the Kalman filter's prediction step, resulting in tighter ELBO bounds. Furthermore, in (Campbell et al., 2021) a method for online state estimation in state-space models (SSMs) was presented. The model operates by utilizing backward decompositions of the joint posterior distribution and Bellman-type recursions to maintain constant update costs.

### 2.3 Neural ODEs

Neural Ordinary Differential Equations (Neural ODEs) (Chen et al., 2018) represent a significant advancement in deep learning by integrating differential equations and neural networks. They provide a flexible framework for modelling complex dynamical systems, treating neural networks as continuous processes. Recent research in this field has made remarkable progress, showcasing the effectiveness of neural ODEs in various applications, especially for modelling irregularly sampled time series and partially observed data such as ODE-RNN, Latent-ODE, NJ-ODE and GRU-ODE models (Rubanova et al., 2019; Herrera et al., 2020; De Brouwer et al., 2019; Abushaqra et al., 2024). These models leverage the ODE's capability to provide continuous hidden states, unlike the traditional discrete approach used in traditional sequence models such as RNN.

In recent years, several models based on Differential Equations (DE) have been introduced to address different issues, including modifying hidden trajectories (Kidger et al., 2020; Morrill et al., 2021; Schirmer et al., 2022) or reducing computational overhead (Habiba & Pearlmutter, 2020). However, these models have typically been evaluated and applied in batch learning and fixed datasets; hence, we do not consider them in this work. Despite the progress in applying neural ODEs to various domains, a critical gap exists in their application to streaming time series data. Existing works primarily focus on batch learning and fixed datasets, lacking exploration of the dynamic and evolving nature of streaming data. The intricacies of adapting neural ODEs to real-time processing of continuously changing data remain unexplored. Our research aims to bridge this gap by introducing ODEStream, a novel framework for continual learning in streaming time series, effectively utilising the strengths of neural ODEs in this challenging context. Moreover, while a recent work (Giannone et al., 2020) proposed real-time classification from event-camera streams using neural ODEs (INODE), its focus on high-frequency event data and real-time classification does not comprehensively address the broader spectrum of challenges associated with streaming time series prediction, which we aim to tackle in this study.

## 3 Preliminaries

In this section, we introduce the fundamental principles of ODEs-based models and ODE solvers, exploring the theory behind neural ODEs and their role in representing irregular time series.

**ODE (Ordinary Differential Equation):** An ODE is a mathematical equation that describes the rate of change of a variable with respect to an independent variable, typically time. In the context of time series data, ODEs represent the dynamics and interactions between variables over time. A simple example of an ODE is: $\frac{dx(t)}{dt} = f(x(t), t)$, where $x(t)$ denotes the state of the system at time $t$, and $f(x(t), t)$ specifies the rate of change at that time.

**ODE Solvers:** Solving an ODE involves determining the function $x(t)$ that satisfies the given differential equation. ODE solvers are numerical methods that approximate the solution over a defined time interval by numerically integrating the ODE using discrete time steps. Starting from an initial condition $x(t_0)$, an ODE solver estimates the values of $x(t)$ at subsequent time points.

**Neural ODEs:** Neural ODEs extend traditional ODEs by parameterise the function $f(x(t), t)$ with a neural network. This allows for modelling complex, continuous-time dynamics in a data-driven manner. The

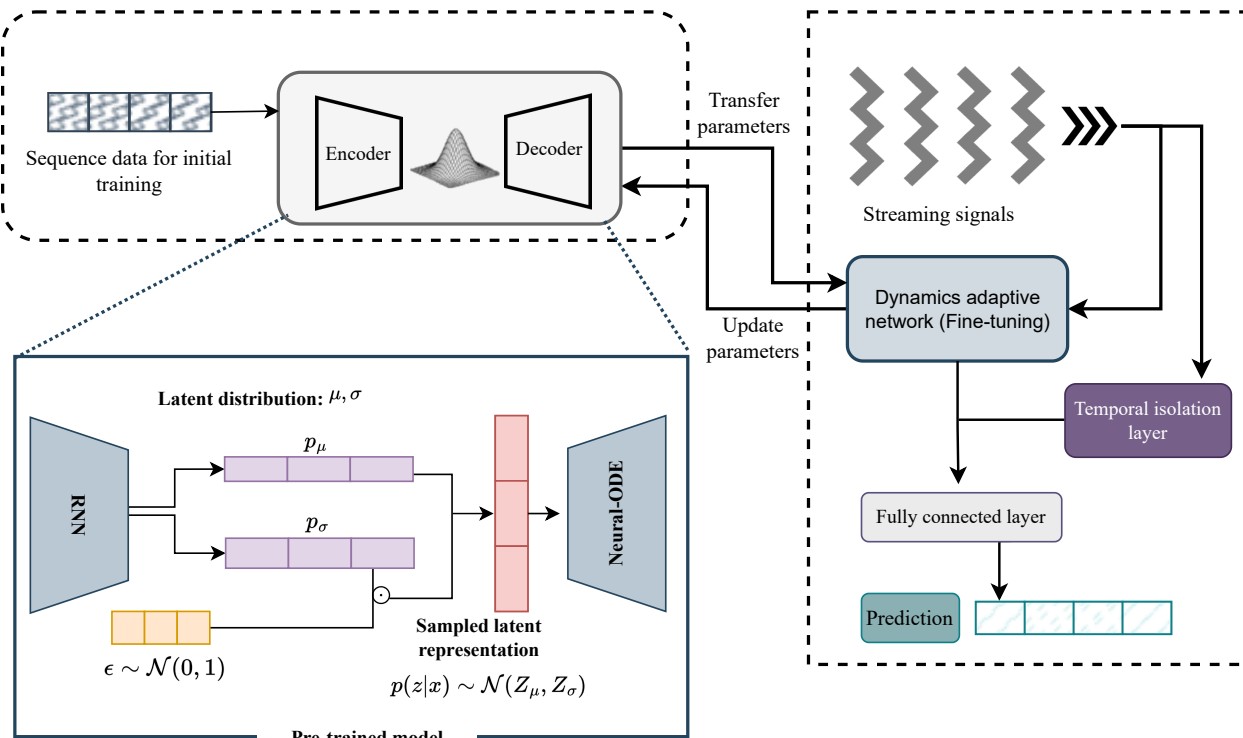

Figure 1: **ODEStream Framework:** which leverages Neural ODEs to encode prior knowledge during the initial training phase. Subsequently, the learned model parameters are transferred and complemented by the temporal isolation layer for continuous online learning. In real-time, streaming data samples are used to predict future values, while the model simultaneously updates its parameters to adapt to newly observed data.

formulation of a neural ODE is:

$$\frac{dh(t)}{d_t} = f_\theta(h(t), t), \quad \text{where} \quad h(t_0) = h_0 \tag{1}$$

$$h_0, \ldots, h_n = \text{ODESolve}(f_\theta, h_0, (t_0, \ldots, t_n)), \tag{2}$$

where $f_\theta$ is a neural network function with parameters $\theta$, and $h(t)$ represents the system's state at time $t$. Given the function $f_\theta$ and an initial condition $h_0$, the ODESolve function computes the values of $h$ at the sequence of time points $t_0, \ldots, t_n$.

## 4 Methodology

**Problem statement:** Let us consider a dynamic environment where streaming time series data is continuously generated as $X = \{x_1, x_2, \ldots, x_t, \ldots\}$ with observations arriving in sequence over time. The goal is to develop a continual learning system for time series forecasting that adapts to the evolving nature of the data stream. Each observation $x_t$ represents the input at time $t$ for univariate data, while $x_t$ represents a vector of inputs at time $t$ for multivariate data, such as $x_t = [x_{t1}, x_{t2}, ..., x_{td}]$, where $x_{td}$ is the value of the $d^{th}$ feature at time $t$. The objective is to predict the future values of the time series. The challenge lies in the continual learning setting, where the model must adapt and learn from the continuously incoming data without access to past observations.

### 4.1 Neural ODEs for Continual Learning

Our goal is to design a training method for memory-free continual learning models. In the context of online learning, we aim for our model to keep learning and adapting as new data arrives without forgetting previously learned knowledge. Traditionally, complex models with buffers are used to store a subset of past data to tackle this issue. As discussed, time series continual learning studies have shown that memory-based approaches often achieve the best performance, particularly in classification tasks. These models continuously monitor incoming data and retrain on both old and new samples to maintain accuracy in dynamic environments, as demonstrated by Fekri et al. (2021). However, When these models are applied to continuous data streams, particularly those with gradually evolving distributions, they face significant challenges. The main issue is that the model only has one opportunity to train on each data sequence, unlike batch learning. Furthermore, in the presence of evolving temporal patterns, it remains unclear how the information stored in the buffer can contribute to future prediction tasks.

After studying the neural ODEs method, we found that it can provide a foundational network for online learning due to the following characteristics (Chen et al., 2018): (*i*) **Continuous-depth representation:** Neural ODEs allow the modelling of flexible continuous-depth neural networks. Unlike traditional discrete-depth neural networks, neural ODEs define the network architecture as a continuous function, making them suitable for tasks where the depth of the network is not predefined. This means they can adapt their depth and complexity as new data points become available. (*ii*) **Memory efficiency:** Neural ODEs do not require storing a large number of intermediate activations, which is valuable when processing streaming data with limited memory resources. (*iii*) **Handling irregular sampling:** Neural ODEs can handle irregularly sampled time series data, which is common in streaming scenarios where data arrive at uneven time intervals or when the features have different temporal representations. This flexibility ensures that the model can adapt to the timing of incoming data points.

### 4.2 Overview of ODEStream

We proposed a streamlined model for continual learning and forecasting of time series, consisting of two main phases. (*i*) an offline learning phase, commonly referred to as model warm-up, designed to initialise the model parameters. (*ii*) online learning and forecasting, which generate predictions based on new sequence flow while adopting and learning from the new changes in the data. The first phase is instrumental in obtaining prior knowledge of the available time sequences. For this part, the model utilises variational autoencoders to capture the distribution, and temporal dependencies of historical data, and an ODE is employed to model evolving dynamics. The trained model is then transitioned to online learning, functioning as a dynamic adaptive network for continuous analysis of streaming sequences. In this primary phase, we leverage pre-trained Neural ODE weights. The model continuously adapts to changes in the data distribution by updating its understanding of the underlying dynamics in real time. As shown in Figure 1, It is noteworthy that the model framework lacks external memory for storing samples, and it does not include a decision node or threshold values to control the flow of the learning process. In the following sections, we provide a detailed description of the model framework.

### 4.3 Obtaining Prior Knowledge (Model warm-up)

Online learning is more complex than batch learning, where the model learns feature representations from sample data and applies them to unseen data. The primary objective of the streaming model is to forecast future values using specific horizontal windows over an extended period. Since the data is unavailable during training, the model does not have the opportunity to process and learn from the sequences over many iterations. In this case, when the data distribution eventually changes, and the sequences evolve over time, the learned features from former historical sequences may no longer be helpful in predicting future values. In general, during the initial training phase of a continual learning model, often referred to as the warm-up phase, the model is pre-trained on historical data from the entire previous period before online learning commences. However, as discussed earlier, this data may have already changed or may change in the near future, so whatever the pre-trained model learns may not remain very relevant in the future. Therefore, relying solely on past data may not fully capture the evolving nature of the streaming sequences.

For the ODEStream model warming-up phase, we do not warm up and train the model on a forecasting task with the goal of learning the relationship between historical data and future time steps. Instead, we follow a new approach. Since the prior data contains information about the distribution, dynamics, and temporal dependencies, our focus lies in initialising our model by utilising the available data to understand how this data evolves and how the dynamics change over time. To achieve this, we leverage the variational autoencoders (VAEs) (Liu et al., 2020) model as shown in the model framework in Figure 1. In contrast to traditional autoencoders, where an encoder layer $e$ maps the data $x$ to a latent vector $Z = e(x)$, in VAE, samples are encoded into a distribution $q(Z|X)$ over the latent variable $z$, rather than being transformed into a single latent representation. The main idea behind a VAE involves two key components; first, an encoder encodes the input data $x$ into a latent variable $z$, capturing its representation distribution through a probabilistic approach parametrized with $\theta$ (Kingma & Welling, 2014), next, a decoder reconstructs the original input data using the latent variable $z$.

In detail, the encoder $e$, which parametrised by $\theta$, produces the outputs: the mean $Z_\mu$ and log variance $Z_\sigma$ of a normal distribution, typically assuming a prior $p(z)$ as a standard multivariate Gaussian distribution, denoted as $\mathcal{N}(0, 1)$. The following equation represents the probability function of the Gaussian distribution:

$$q(z|x) = \mathcal{N}(Z_\mu, Z_\sigma), \tag{3}$$

here, $q(z|x)$ is the conditional distribution of the data $x$ given the latent variable $z$, and $\mathcal{N}$ is the Gaussian distribution with mean $Z_\mu$ and standard deviation $Z_\sigma$. The latent variable $z$ is sampled from the reparameterized distribution as:

$$Z = z_\mu + \epsilon \bigodot exp(0.5 \cdot z_\sigma), \tag{4}$$

where $\epsilon$ is a random sample drawn from a standard normal distribution $\mathcal{N}(0, 1)$. This sampled latent variable $z$ is then fed into the decoder, which generates the final output $y$ by expressing p(x|z) as:

$$p(x|z) = \mathcal{N}(\mu_y, \sigma_y^2), \tag{5}$$

where $\mu_y$ and $\sigma_y^2$ are the parameters of the decoder.

To obtain the dynamics and learn how the sequence distribution changes over time, we use neural ODEs. As ODEs can learn the dynamics over time, they are suitable for tracking changes in time series data. The RNN encoder processes the input sequence $x$ to obtain the hidden state $h_0$. The hidden state is then mapped to the mean $Z_\mu$ and log variance $Z_\sigma$ discussed above using a linear layer. The latent variable $z$ is sampled from a Gaussian distribution as: $z_{t0} \sim \mathcal{N}(z_\mu, z_\sigma)$

Given a sequential data sequence $\{x_1, x_2, \cdots, x_T\}$ with a sequence length of $T$, a neural RNN encoder processes each time step $t$ to produce a hidden state $h_t$. The encoder estimates the parameters $Z_\mu$ and $Z_\sigma$ of the variational posterior distribution $q(z_t|x_t)$ for the latent variable $z_t$. Subsequently, a fully connected layer maps the hidden state to $h_0$, and samples $z_{t0}$ using the following equation:

$$z_{t0} \sim q(z_{t0}|x_{t0}, \ldots, x_T; t_0, \ldots, t_T; \theta_p) = \mathcal{N}(z_{t0}|z_{\mu_t}, z_{\sigma_t}) \tag{6}$$

The latent trajectories, which represent the evolution of the latent variable $z_t$ over time, are computed using an ODE solver as Equation 7 (using the Euler method in our case). This solver is employed to model how the latent space changes and adapts as the sequential data is processed using the initial latent state $z_{t0}$. The ODE solver, represented as $\frac{dz}{dt} = f(z, t; \theta_f)$, is a mathematical tool that describes how the latent variable $z$ changes concerning time $t$ and is parameterised by $\theta_f$. Therefore, it captures the dynamic behaviour of the latent space over the entire sequence (the data for the initial training) before the online learning commences.

$$z_{t1}, z_{t2}, \cdots, z_T = ODESolve(z_{t0}, f, \theta_f, t_0, \cdots, t_T) \tag{7}$$

## 4.4 Online Learning

Following the initial training phase, during which the neural ODE model learns from historical data, the model transitions into online learning. In this phase, the trained model is deployed as an adaptive dynamic

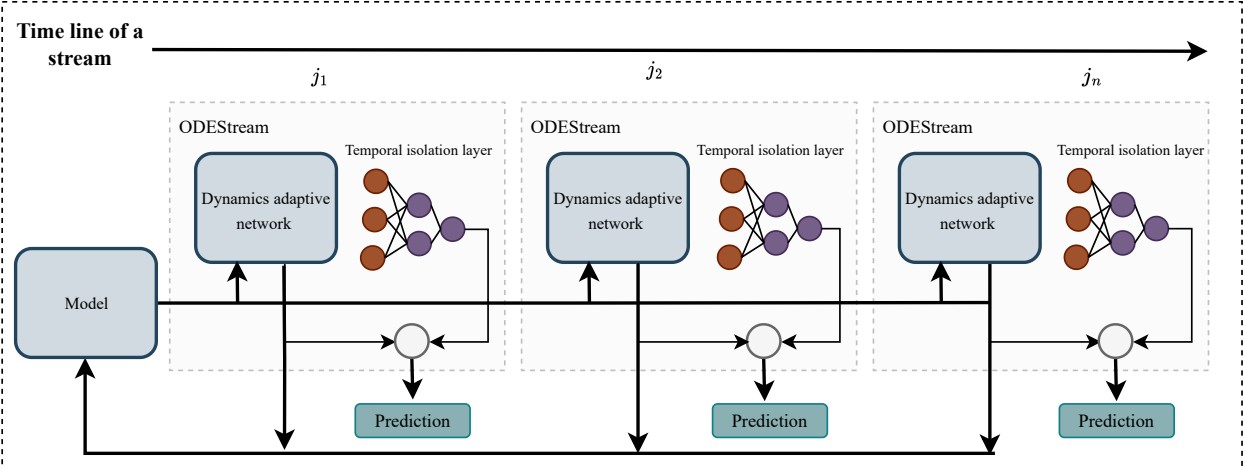

Figure 2: **The continual learning process of ODEStream**. The model processes the incoming observation using the pre-trained VAE-ODE and continuously changes its parameters based on new dynamics.

network, enabling it to continuously process and analyse new streaming sequences as they become available (as illustrated in Figure 2). For simplicity, we denote the sequence $X = \{x_{i+1}, x_{i+2}, x_{i+3}, \dots\}$, as $X = \{x_1, x_2, x_3, \dots, x_j, \dots\}$, where $j$ represents the time point following $i$, assuming $i$ denotes the last data point utilized in the initial training phase.

The main role of the adaptive network is twofold: first, to generate forecasts based on the recent available data $x_j$, and second, to adapt to changes in the data distribution. This adaptation is driven not only by prediction accuracy but also by the evolution of the data distribution, which will be discussed in Section 4.5.

The model architecture is intentionally designed to support a continual understanding of how the underlying data dynamics evolve over time. This achieved by continuously integrating new data sequences through the neural ODE framework, allowing the model to detect and adjust to shifts in data distribution. Specifically, the pre-trained neural ODE weights and structure are reloaded to model the latent space dynamics $z(j)$ of the streaming sample $x_j$, where $j$ denotes the most recent value in the data stream. The latent space representation $z(j)$ is then linearly mapped to a hidden space $(h_j)$, which subsequently maps linearly to the output space $(y_j)$, as defined by:

$$z(j) = \text{NeuralODE}(z_0, j) \tag{8}$$
$$h_j = \text{Linear}(z(j)) \tag{9}$$
$$y_j = \text{Linear}(h_j) \tag{10}$$

Based on the learned dynamics, the model parameters $\theta$ are updated iteratively as $\theta_{\text{new}} = \theta_{\text{old}} - \eta \cdot \nabla_\theta \text{Loss}$.

**Temporal Isolation Layer:** To further enhance the model's adaptability and responsiveness, we incorporated it with a specialised component known as the temporal isolation layer. This neural network serves two purposes: $(i)$ Focus on new information, as it resembles a dynamic "window" into the most recent data samples. Its primary objective is to extract and learn from the unique temporal patterns and features embedded within the recent data. By closely monitoring the look-back windows and forecast horizon of the data, this layer ensures that the model effectively captures emerging trends and patterns. $(ii)$ Avoid historical bias, where, in addition to the focus on new information, we also manage the influence of past patterns learned by the pre-trained model that may have become obsolete or inaccurate. By independently processing and learning from the most recent data samples, this layer reduces the risk of the model being unduly guided by outdated historical data, ensuring its predictions remain relevant and reliable.

The input of this layer is the current stream sequence with a specific look back window (lag) $(x_{j-lag}, \ldots, x_j)$, where $j$ is the most recent time point of a stream. As shown in Equation 11 and 12, the temporal isolation layer $(f)$ generates the latent representation of that sample $h'_j$, and a concatenate layer $(g)$ combines the latent representation of the samples $h'_j$ along with the neural ODE-based representation $h_j$ of that sample to generate the final prediction. $c_j$ is a cell state as we employ an LSTM for this layer

$$h'_j, c_j \Rightarrow f(x_j, h_{j-1}, c_{j-1}) \tag{11}$$

$$h''_j = g(h'_j, h_j) \tag{12}$$

Incorporating the temporal isolation layer allows the model to maintain a delicate balance between preserving valuable insights from historical data and staying attuned to the evolving nature of the data stream. This adaptive approach positions the model as a robust and adaptable tool for real-time forecasting and concept adaptation in dynamic environments.

## 4.5 Model Regularisation

To ensure that the model comprehensively grasps the underlying data distribution, we used Kullback-Leibler (KL) divergence loss as Equation 13 (Csiszár, 1975) to guide the pre-trained model and encourage the continual learning process to adapt to distribution changes. KL divergence measures the disparity between two probability distributions, which facilitates the transformation of data into a latent space while encouraging the latent variables to adhere to a specific distribution. In other words, it ensures that the model comprehensively captures the data distribution, unlike the Mean Squared Error (MSE) loss, which typically focuses solely on optimising output quality. However, as the final output is a future prediction, it is still important to evaluate the prediction performance of the model while learning new dynamics. Hence, a combination of KL and MSE loss was utilised in the continual learning process.

$$\mathcal{L}_{KL} = -0.5 * \sum_{i=1}^{N} 1 + log(Z_{\sigma_i}^2) - Z_{\mu_i}^2 - Z_{\sigma_i}^2 \tag{13}$$

Furthermore, to control overfitting, we used regularisation methods that play a significant role in enhancing the robustness and generalisation of machine learning models and encouraging the development of simpler, more adaptable models. We used L1 regularisation in Equation 14 as an additional protection for enhanced model simplicity. L1 regularisation penalises the absolute values of neural network weights.

$$\mathcal{L}_{L1} = 0.01 * \sum_{i=1}^{N} |z_i| \tag{14}$$

Algorithm 1 outlines the ODEStream online learning algorithm, in summary, the algorithm employs a pre-trained Neural VAE-ODE model within an online learning framework, processing the streaming data sequence $X = \{x_1, x_2, \ldots, x_j\}$ and updating the model parameters as new data becomes available. The ODEStream algorithm operates iteratively for each incoming data sample $x_j$, integrating both the neural ODE representation and the temporal isolation layer to generate predictions and update model parameter. Initially, the model calculates a latent representation $z(j)$ of the streaming sample using the NeuralODE function, followed by a linear mapping to hidden and output spaces $h_j$. Simultaneously, the temporal isolation layer generates an independent latent representation $h'_j$ for the new sample, isolating recent patterns and minimizing reliance on potentially outdated historical data. At each time step, the algorithm combines the representations $h_j$ and $h'_j$ using a concatenation layer to produce an updated hidden state $h''_j$, yielding the final prediction $y_{j+1}$. The model parameters $\theta$ are then updated via gradient descent, optimizing with a composite loss function incorporating mean squared error (MSE), KL-divergence $\mathcal{L}_{KL}$, and L1 regularization $\mathcal{L}_{L1}$. This procedure ensures that the model adapts to new data patterns while preserving relevant insights from previous training.

---

**Algorithm 1: ODEStream - Online Learning**

---

**1** **Input:** Pre-trained NeuralODE model, streaming sequence $X = \{x_1, x_2, \ldots, x_j\}$, previously learned parameters $\theta$

**2** **for each time step** $j$ **in** $X = \{x_1, x_2, \ldots, x_j\}$ **do:**

**3**     $z(j) = \textbf{NeuralODE}(z_0, j)$

**4**     $h_j = \textbf{Linear}(z(j))$

**5**     $h'_j, c_j \leftarrow f(x_j, h_{j-1}, c_{j-1})$

**6**     $h''_j = g(h'_j, h_j)$

**7**     $y_{j+1} = \textbf{Linear}(h''_j)$

**8** **Update model parameters**

**9**     $\mathcal{L}_{\textbf{KL}} = -0.5 \sum_{i=1}^{N} \left( 1 + \log(Z^2_{\sigma_i}) - Z^2_{\mu_i} - Z^2_{\sigma_i} \right)$

**10**     $\mathcal{L}_{\textbf{L1}} = 0.01 \sum_{i=1}^{N} |z_i|$

**11**     $\textbf{Loss} = \mathcal{L}_{\textbf{MSE}} + \mathcal{L}_{\textbf{KL}} + \mathcal{L}_{\textbf{L1}}$

**12**     $\theta_{\textbf{new}} = \theta_{\textbf{old}} - \eta \cdot \nabla_\theta \textbf{Loss}$

**13** **end for**

**14** **Output: Updated model parameters** $\theta$ **and final prediction** $y_{j+1}$

---

## 5 Experiments

In our experiments, we aim to address three primary questions that collectively offer a comprehensive assessment of the model's performance:

- Does ODEStream outperform other recent methods, including ER and online forecasting techniques?

- How adaptable is the model to incoming data? How does it evolve over extended periods of streaming data, and how does concept drift in the data influence its performance?

- Will ODEStream preserve its performance even when applied to irregular and sparse streaming data?

### 5.1 Experimental Setup

#### 5.1.1 Datasets

To evaluate the performance of our proposed model, we employed several benchmark real-world forecasting datasets. We used (1) Electricity Consumption Load (ECL) [1] which is a public dataset describes the electricity consumption (Kwh) of 321 clients. (2) Electricity Transformer Temperature (ETT) (Zhou et al., 2021)[2] which is multivariate time series records two years of data, focusing on predicting the electrical transformers' oil temperature based on load capacity. The dataset includes six features, and we utilised both the available hourly (h) dataset (two sites separately) and minute (m) dataset. (3) Weather Data (WTH)[3] which describes local climatological conditions for various US sites. It includes 11 climate features and the target value (a wet-bulb temperature).

#### 5.1.2 Baseline

In addition to a non-continual learning baseline using a basic RNN layer, we evaluated a set of baselines for continual learning time series forecasting. As FSNET (Pham et al., 2023) is the most recent state-of-the-art model, we consider it as our main baseline along with its variant naive that showed better performance for one step ahead forecasting. We also include a basic buffer-based model (Chaudhry et al., 2019b); Experience

---

[1] https://archive.ics.uci.edu/ml/datasets/ElectricityLoad Diagrams20112014

[2] https://github.com/zhouhaoyi/ETDataset

[3] https://www.ncei.noaa.gov/ data/local-climatological-data

Replay (ER). ER enhances the OnlineTCN by incorporating an episodic memory that preserves past samples, subsequently interleaving them during the learning of more recent ones. Additionally, we consider a recent variant of ER, Dark experience replay DER++ (Buzzega et al., 2020), that Enhances the conventional ER by introducing a knowledge distillation loss on the previous logits.

### 5.1.3 Setup and Experimental details

In the encoder part of the VAE, we utilize RNN model consisting of one GRU layer with a hidden dimension of 64, followed by a linear layer that outputs a latent dimension of 64. The decoder comprises three main components: a Neural-ODE that initializes the latent state $z$, a linear layer that maps the latent state to the hidden dimension of 64, and a second linear layer that maps the hidden representation to the final output. The dynamics of the system are integrated over time using the Euler method, with the step size determined based on the time duration from the initial time $t_0$ to the final time $t_1$, constrained by a maximum step size of 0.05.

For all the experiments, we applied non-shuffled splitting to divide the data into a training set (25% ) for initialising the model (warm-up training) and a streaming dataset (75%) for online testing and training. We rescaled (normalised) the features using a standard normal distribution that fits on the training set only. We used a look-back window of 24 for our model and the baselines, while the learning rate was set to 0.001, and we employed the Adam optimiser for model optimisation. During the warm-up phase, we utilized a batch size of 64 and implemented early stopping after 10 epochs. To simulate a streaming environment for online learning, the batch size and epoch were set to one.

Table 1: Cumulative MSE results of several datasets using ODEstream and the baseline models for different tasks, including: ( • ) univariate predict univariate, (*) multivariate predict univariate, ($\frown$) multivariate predict multivariate.

| Method | ECL • | ECL$\frown$ | ETTh1* | ETTh1$\frown$ | ETTh2* | ETTh2$\frown$ | ETTm1* | ETTm1$\frown$ | WTH* | WTH$\frown$ |
|---|---|---|---|---|---|---|---|---|---|---|
| RNN | 0.576 | 33.84 | 43.36 | 1.2653 | 37.35 | 6.3711 | 41.08 | 0.470 | 0.1636 | 0.4616 |
| ER | 2.8142 | 2.8359 | 1.9785 | 0.2349 | 6.7558 | 0.5044 | 3.055 | 0.082 | 0.3138 | 0.1788 |
| DER++ | 2.8107 | **2.81** | 1.9712 | 0.24 | 6.738 | 0.5042 | 3.0467 | **0.0808** | 0.3097 | **0.1717** |
| FSnetNaive | 2.9943 | 3.0533 | 2.001 | 0.2296 | 6.7749 | 0.5033 | 3.0595 | 0.1143 | 0.3843 | 0.2462 |
| FSnet | 2.8048 | 3.6002 | 1.9342 | 0.2814 | 6.681 | 0.4388 | 3.0467 | 0.0866 | 0.3096 | 0.1633 |
| ODEStream | **0.1173** | 4.095 | **0.0594** | **0.105** | **0.164** | **0.1879** | **0.0625** | 0.2178 | **0.0441** | 0.222 |

## 5.2 Evaluation Results

The results of the ODEStream performance compared to the baseline models are presented in Table 1. Our experimental encompassed multiple tasks, including univariate and multivariate forecasting tasks, denoted as ( • ) and (*) respectively in the table, along with a multivariate-multivariate forecasting task (represented as $\frown$) where the model predicts the future target value with the future feature list as applied in (Pham et al., 2023).

In the case of the ECL dataset, which is naturally univariate data, we conducted univariate forecasting by analysing the historical data of a specific variable (a single client). All other datasets are multivariate data, where we leverage multiple features to predict future values of a single target variable as multivariate fore-casting tasks. For all datasets, we used all available features to predict a set of future values for these features for multivariate-multivariate forecasting. It is important to note that while reviewing the baseline models and the proposed model described in (Pham et al., 2023), we observed that the reported performance metrics were primarily for the multivariate-multivariate forecasting task where a multivariate predict multivariate. However, we extended our analysis to include the prediction performance for other tasks as well. Regarding the non-continual learning baseline (RNN model) the performance is suboptimal due to the fact that these models are not designed for continual learning and are unsuitable for handling streaming environments effectively. Upon closer examination, the performance shown by the other baseline models was not as high as anticipated in tasks other than multivariate-multivariate forecasting. For this task, the performance of our method closely approximated that of the baseline models. However, we achieved a substantial enhancement

in the performance of univariate forecasting tasks when compared to the baseline models. Figure 3 and 4 illustrate the performance of ODEStream, FSNet and DER++ over a massive period of streaming, and we can see how well ODEStream and the baseline models adapt to gradual changes in the datasets through continuous learning. It is evident that FSNet and DER++ struggle to adapt to all the data changes over an extended period, resulting in predictions that diverge significantly from ground truth values for most of the dataset by the end of the streaming sequence. Conversely, ODEStream continuously adapts to new dynamics across the entire duration, which explains why it achieved lower cumulative MSE values. It is also worth mentioning that DER++ outperforms in multivariate forecasting due to the complex and non-stationary nature of the relationships between different variables. DER++ leverages a replay buffer that stores a small subset of past data points, allowing the model to rehearse past experiences. This mechanism helps maintain performance on previous data distributions while adapting to new patterns, which is crucial for handling the intricacies of multivariate time series data.

In addition to the main experiment, Table 2 presents the results of the multi-horizon predictions for 7 and 24 steps across five datasets. The performance remains stable over longer prediction horizons, with no significant degradation in accuracy. Notably, even at 7 and 24 steps, the model consistently outperforms the baseline models from previous experiments.

Table 2: Multi-horizon prediction results.

| steps | ECL | ETTH1 | ETTH2 | ETTM1 | WTH |
|---|---|---|---|---|---|
| 1 | 0.1173 | 0.0594 | 0.164 | 0.0625 | 0.0441 |
| 7 | 0.403 | 0.1006 | 0.4009 | 0.5997 | 0.1089 |
| 24 | 1.003 | 0.1909 | 0.3751 | 0.09186 | 0.133 |

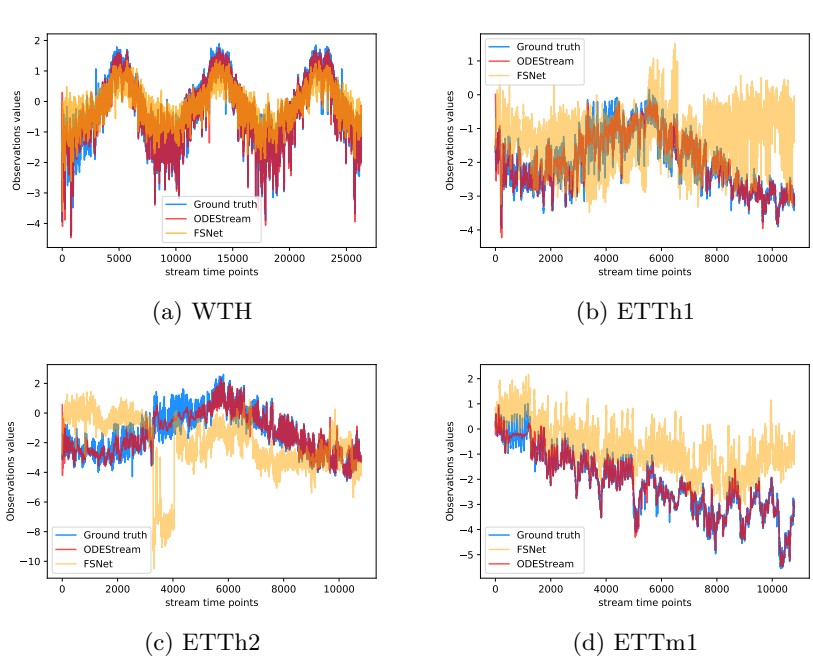

(a) WTH  (b) ETTh1

(c) ETTh2  (d) ETTm1

Figure 3: ODEStream against FSNet: Adapting dynamics evolution for the entire stream sequence of several datasets.

## 5.3 Performance on Irregular Data

In this section, we assess the model performance on irregular sparse data, essentially evaluating its ability to continuously learn from samples with uneven time intervals. To simulate irregular sparse data, we apply a cut-out function to the benchmark dataset, following the work in (Rubanova et al., 2019). Specifically, we

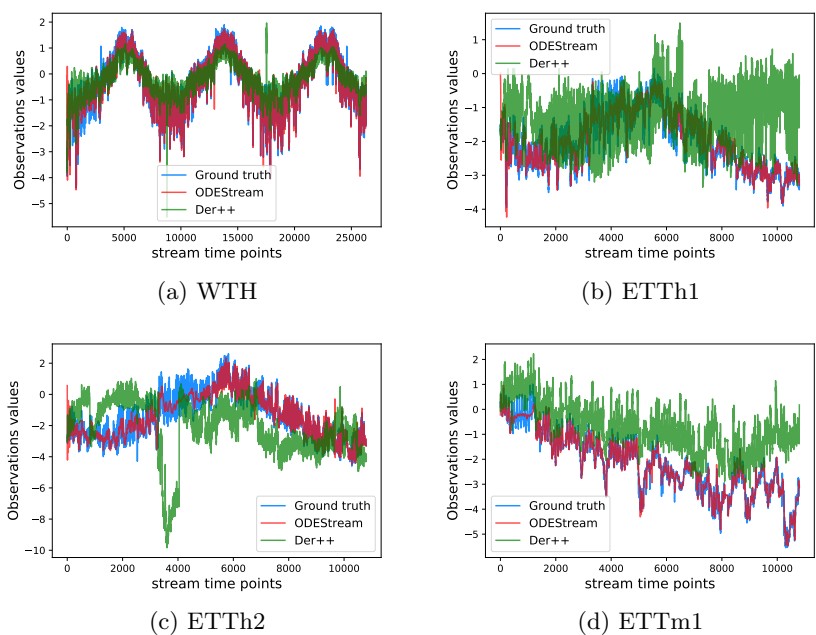

Figure 4: ODEStream against DER++: Adapting dynamics evolution for the entire stream sequence of several datasets.

remove 30% of the data points for each attribute in the sequences. Thus we ensure that there is no fixed time interval between observations, and not all samples are fully observed at every time point. Figure 5 illustrates the model performance with regularly sampled data compared to irregularly sampled data. We observe that the error drops by an average of less than 0.020 score. While most cases experience a minor increase in error, this could imply that the model adapts well to irregularities in the sampling pattern.

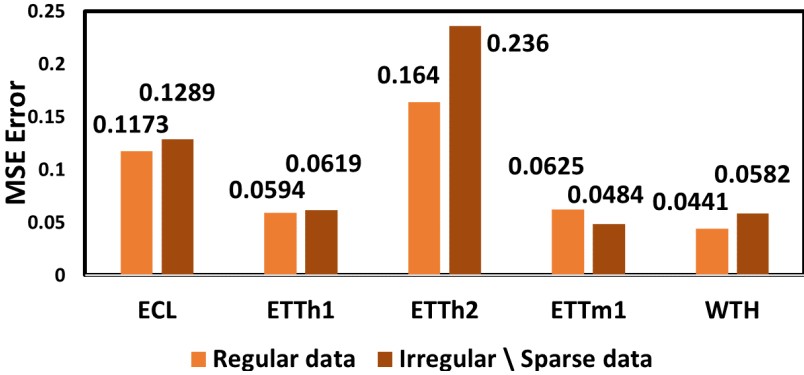

Figure 5: Model performance on irregularly sampled data.

## 5.4 ODEStream Adaptability

In order to assess the robustness of our proposed model in the presence of concept drift, we employed the Adaptive Windowing (ADWIN) method for concept drift detection to the testing set of all datasets under consideration and evaluated the performance of our model in adapting to these drifts. Figure 6 provided a visual representation of how our model responded to changing data patterns compared to a baseline model. The red vertical lines represent a drift in the data detected by ADWIN. The predictions generated by

ODEStream shed light on the model's ability to maintain effectiveness and accuracy even in dynamic and evolving data environments.

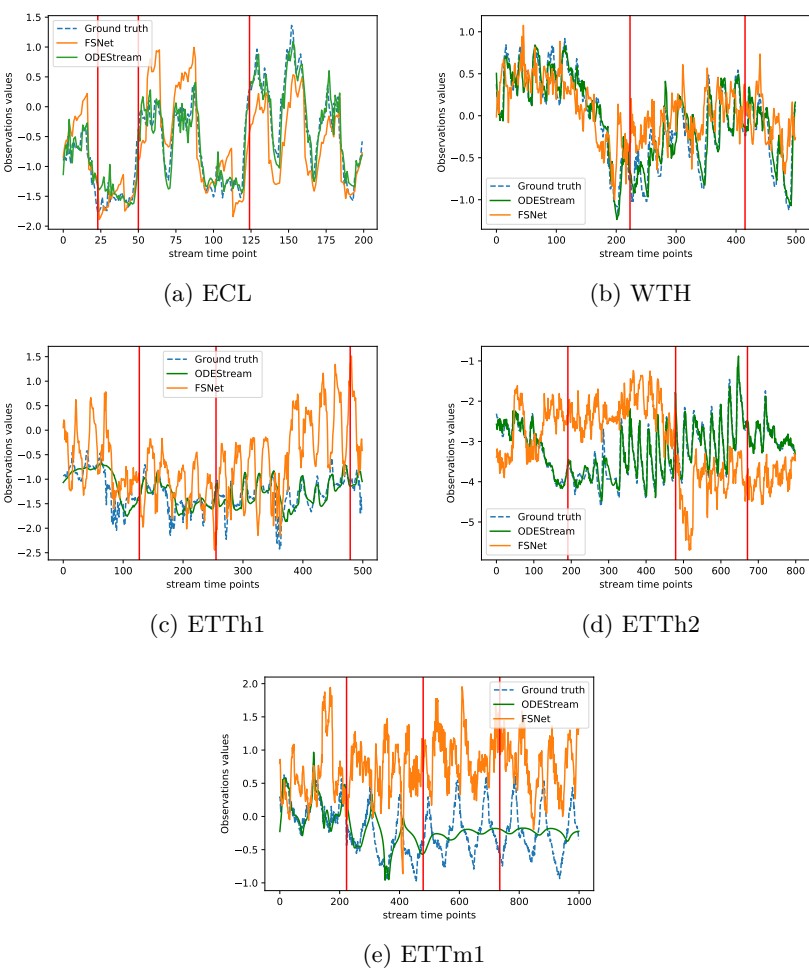

Figure 6: Predictions by ODEStream compared to the baseline model FSNet amid concept drift (indicated by red vertical line).

## 6 Ablation study

For the ablation study, we systematically examined the performance of three model configurations: (1) the full framework with both the temporal isolation layer and the combination of MSE and KL loss for convergence, (2) the model without the temporal isolation layer, but still utilising the MSE and KL loss, and (3) the model without the temporal isolation layer, with only the MSE loss for convergence. By comparing these configurations, we gained insights into the effectiveness and necessity of the temporal isolation layer and the impact of different loss functions on the model's performance in handling concept drift and dynamic time series data. The results of the ablation study are shown in Table 3, demonstrating the significant effectiveness of the temporal isolation layer, which improved performance by more than 50% on average across all datasets.

Table 3: Cumulative MSE results of the ablation study with/without a temporal isolation layer and with using different loss functions (1) MSE, (2) MSE-KL.

| Method | ECL | ETTH1 | ETTH2 | ETTM1 | WTH |
|---|---|---|---|---|---|
| W/O TIL$_1$ | 0.299 | 0.127 | 0.372 | 0.10 | 0.200 |
| W/O TIL$_2$ | 0.369 | 0.084 | 0.347 | 0.098 | 0.178 |
| W/ TIL$_1$ | 0.119 | **0.054** | 0.176 | 0.09 | 0.060 |
| ODEStream | **0.117** | 0.059 | **0.164** | **0.062** | **0.044** |

## 7 Model Robustness

### 7.1 MSE Convergence

Figure 7 illustrates the convergence of MSE values during streaming prediction on all datasets for both ODEStream and FSNet. It is evident that ODEStream maintains lower MSE values even after an extended period of time. This indicates the model's ability to continuously learn from new observations as they become available over time. Additionally, in certain cases, ODEStream's MSE values decrease over time, in contrast to the baseline model, where the error values tend to increase with time.

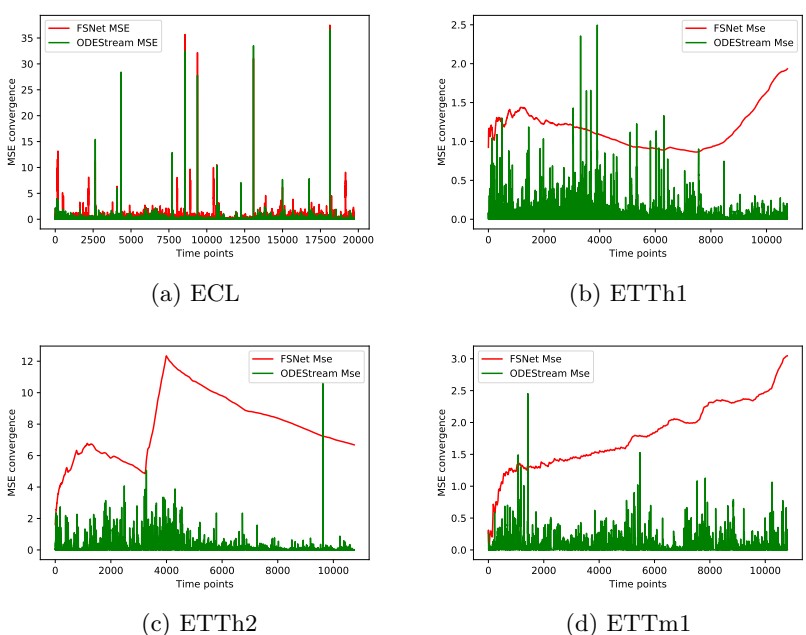

Figure 7: MSE convergence using ODEStream compared to FSNet for several datasets.

### 7.2 Computational Resources

In Figure 8, we show the required resources in terms of time and memory. In the first three plots, we present the required time needed to process a stream of sequences for each dataset. We have added the FSnet required time as a reference to show the improvement and provide insights into the efficiency of ODEStream. For example, Figure 8a shows the average time per second required to process one stream sequence. ODEStream requires 88% less time on average compared with the baseline model. In Figure 8d, the average memory usage in megabytes is calculated using the "*psutil*" library for each process.

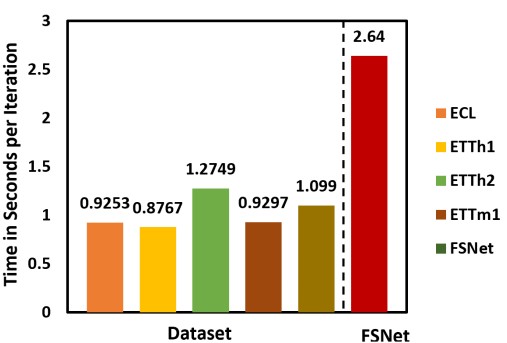

(a) Execution time in seconds per one sequence.

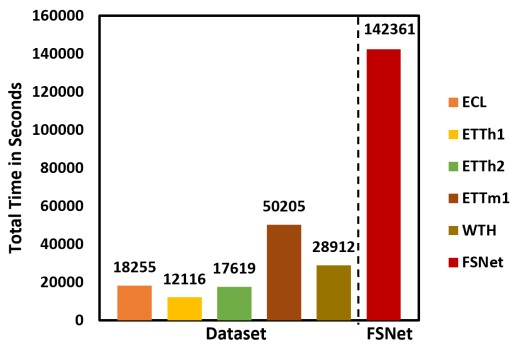

(b) Execution time in seconds for the whole stream-
ing period.

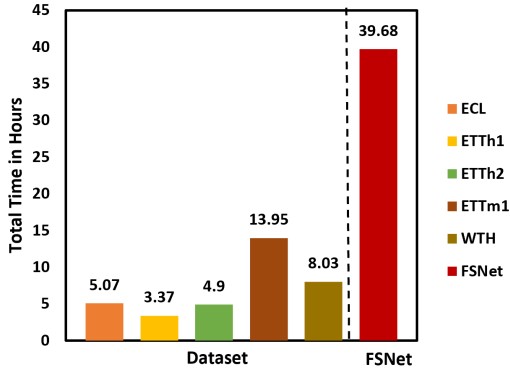

(c) Execution time in hours for the whole streaming
period.

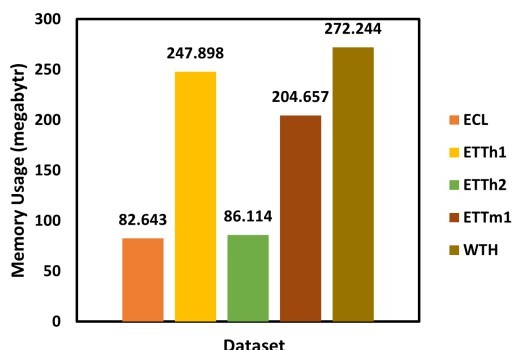

(d) The average memory usage for each sequence
processing.

Figure 8: ODEStream computational resources in terms of time and memory. FSNet is used as a reference.

## 8 Conclusion

In this article, we introduce a new approach to buffer-free online forecasting from streaming time series data, addressing the critical need for real-time adaptability to changes. The proposed model, ODEStream, leverages the capabilities of neural ordinary differential equations (ODEs) to dynamically adapt to irregular data, and concept drift, eliminating the necessity for complex frameworks. Through continuous learning and incorporating a temporal isolation layer, ODEStream stands out in processing streaming data with irreg- ular timestamps and efficiently handles temporal dependencies and changes. Our evaluation of real-world benchmark time series datasets in a streaming environment demonstrates that ODEStream consistently out- performs a range of online learning and streaming analysis baselines, confirming its effectiveness in providing accurate predictions over extended periods while reducing the challenges associated with stream time series data processing. This research marks a significant stride in advancing the capabilities of online forecasting, offering a promising solution for real-time analysis of dynamic time series data. Looking ahead, our future work aims to extend the applicability of the proposed methodology to various time series continual learning scenarios. This includes exploring task incremental learning scenarios where a sequence of $m$ tasks arrive, each task consisting of N instances of labelled time series data. For each task, ODEStream will be employed to learn a solver for each task with no access to the data of previous tasks.

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
