# OpenReview forum: "ODEStream: A Buffer-Free Online Learning Framework with ODE-based Adaptor for Streaming Time Series Forecasting"
_TMLR — Accepted by TMLR_

### Review · Reviewer_UhMa · 2024-10-07

**Summary Of Contributions:**

This paper introduces ODEStream, a novel model for online learning and forecasting in streaming time series data, emphasizing adaptability without the need for complex frameworks. Unlike traditional methods that rely on preserving historical data, ODEStream focuses on learning dynamic patterns and distributions in continuous time series, leveraging neural Ordinary Differential Equation (ODE) models to handle irregularly sampled data. It presents a streamlined, buffer-free approach that simplifies training by eliminating the need for decision-making processes like thresholding or buffering. Comprehensive evaluations show that ODEStream effectively adapts to data drifts and irregularities, maintaining stable performance over long streaming periods and outperforming state-of-the-art continual learning models.

**Audience:**

Yes

**Claims And Evidence:**

Yes

**Requested Changes:**

See above

**Strengths And Weaknesses:**

**Strengths:**
- Comprehensive literature review.
- Tackles a very important and interesting problem.
- Delivers impressive experimental results.
- Experiments cover performance on both regular and irregular time series, as well as robustness and computational efficiency.
- Being able to deal with irregular time series is indeed very important for real life applications.

**Weaknesses:**
- While reviewing related works like FSNet, it would be beneficial to test the proposed method on Traffic and other benchmark datasets.
- The experiments seem to focus only on one-step ahead forecasting. It would be valuable to include multi-horizon predictions, or is there a reason this wasn’t explored?

**Questions:**
- Can the authors clarify if they re-ran the experiments for the baselines or changed the experimental settings? If so, why was this approach taken instead of directly using the reported baseline results?
- For the univariate case, did the authors carefully perform hyperparameter tuning for the baselines? If so, can they report the details in the appendix?

**Minor Question:**
- For long streaming scenarios, while I understand the proposed method negates the need for retraining, would it still make sense to discard past observations and retrain the model using more recent data, especially when a large amount has been collected over time?

---

> ### Author Response · Authors · 2024-10-27
> **Discussing the Weaknesses and Questions**
>
> We sincerely appreciate your thoughtful feedback and suggestions.  Below, we provide detailed responses to your points, along with our proposed changes to the manuscript.
>
> ## Weaknesses:
>
> ### 1. Testing on Traffic dataset:
> Thank you for bringing this to our attention. We have incorporated most of the real-world datasets used for FSNet, and their paper provides detailed descriptions of the target variables, sequence lengths, and dimensionality. However, there is limited information about the specific time period and exact areas covered for the Traffic dataset, and the final processed data was not available in the GitHub repository. This made it challenging to replicate their exact experiments. However, If you believe that including results on the Traffic dataset is essential, we are happy to include our own Traffic dataset and conduct experiments on ODEStream and all baselines for the final version of the paper.
>
>
>
> ### 2. Multi-Horizon predictions:
> There was no specific reason for excluding multi-horizon predictions in our initial experiments. Our focus was primarily on three key aspects: evaluating the model’s ability to maintain performance without retraining on old data, assessing how the model adapts to data drift, and analyzing its behavior with irregular time series, in addition to evaluating resource requirements in terms of time and memory. However, we appreciate your suggestion and are currently running experiments on multi-step forecasting. We will include these results in the final version of the paper and will share them in a separate comment once completed.
>
>
> ## Questions:
> ### 1.  Baseline experiments:
> We re-ran the experiments for the baselines using slightly modified settings. The original work utilized a look-back window of 60; however, we discovered that accurate predictions could still be generated with a look-back window of 24. This adjustment was necessary due to resource limitations on the machine used for our experiments. To ensure consistency across our experiments, we set the look-back window to 24 for our model as well as all the baselines. This change should be beneficial, as it reduces computational resource usage while maintaining high performance.
>
> ### 2. Hyperparameter tuning for the univariate case:
> We used the main parameters from the original baseline implementations, as they were noted to yield optimal performance. However, we understand the concerns about the lower performance for these models, particularly since univariate data experiments were not included in the previous work. As we can see from our experiments, baseline models struggled with adapting to data drift, as demonstrated in Figures 3, 4, and 6. We believe that a drift in a single variable in a univariate case can have a substantial effect on predictions, while in multivariate settings, drifts in individual variables tend to have less impact. This could be a key reason for the lower performance observed in the univariate experiments using the baseline models. We will include more details on hyperparameters in the appendix and provide additional insights in the results section.
>
> ### 3. Long streaming scenarios:
> Regarding your last question, we would like to highlight that
> ODEStream does not entirely ignore older sequences. Instead, it leverages them to capture the dynamics of the features, which is key to preventing catastrophic forgetting. Traditional models typically require retraining on both new and old data to avoid forgetting, but ODEStream utilizes old sequences to understand changes over time, which helps in handling concept drift. This design maintains high performance over long streaming periods without being overwhelmed by historical data.
>
>
> Thank you once again for your valuable feedback, we are happy to clarify further questions.

---

> > ### Author Response · Authors · 2025-01-17
> > **Multi-horizon predictions**
> >
> > Dear Reviewer, the experiments evaluating multi-horizon predictions (specifically for 7 and 24 steps ahead) is summarized in the table below.
> >
> > Notably, there is no significant drop in the performance of the model, and it consistently outperforms the baseline models compared in the previous experiment.
> >
> > | Steps | ECL      | ETTh1    | ETTh2    | ETTm1    | WTH      |
> > |-------|----------|----------|----------|----------|----------|
> > | 1     | 0.1173   | 0.0594   | 0.164    | 0.0625   | 0.0441   |
> > | 7     | 0.403    | 0.1006   | 0.4009   | 0.5997   | 0.1089   |
> > | 24    | 1.003    | 0.1909   | 0.3751   | 0.09186  | 0.133    |

---

### Review · Reviewer_G8NS · 2024-10-17

**Summary Of Contributions:**

In this paper, the authors introduce ODEStream, a continual learning method designed streaming time series. The proposed method is a buffer-free approach to continual learning, where neural ODEs are used to also allow the model handle non-constant data sampling (which is very common in the real world)

**Audience:**

Yes

**Claims And Evidence:**

Yes

**Requested Changes:**

1.In the VAE literature, $ q(z \mid x)$ is commonly used to denote the encoder while the $p$ is used to denote the prior and the decoder. Thus, it would be better the authors follow this convention. *Not critical*

2. Equations 3 and 5 seem like the same exact thing. *Not critical*

3. Adding equations and algorithm table to make the paper easier to follow and implement. *Critical*

4. Reference works for deep-learning based methods for streaming time series data. *Critical*

5. Experimental details. *Critical*

6. A non-continual learning baseline. *Critical*

7. It is not clear what lift the neural ODE is providing. It would be great if a version parameterized by an RNN was included in the ablation study. *Critical*

**Strengths And Weaknesses:**

# Strengths

Firstly, I very much love the simplicity of the proposed model. The lack of a buffer makes it significantly easier to deploy this in practice as one does not have to tune the buffer configuration. Moreover, I am *very* impressed with the results of the proposed model. The experiments are varied and are all applied on real world data.

# Weaknesses

While I like the simplicity of the proposed approach, there are major weaknesses. Firstly, I think the writing needs substantial work. For starters, while I appreciate how the method is described in plain english, it makes it very hard to decipher what is going on. As an example, in the VAE section, the authors spend a paragraph describing the evidence lower bound (ELBO) in plain english, where it would have been significantly more clearer to just write the ELBO. Moreover, it would make it substantially clearer that the authors are only inferring the initial condition and that the subsequent latent states are generated deterministically. It took me quite a few reads of this section to catch this, as only having a stochastic initial condition in a sequential VAE model is generally non-standard, though see [2] for another work that does this. Also, the authors switch between $x$ and $y$ to denote the observations, which makes things confusing.

The lack of details is extended into the online learning section, as it is not clear at all what is going on. A couple of equations and an algorithm would go quite a long way explaining what is going on here, especially with the inclusion of the temporal isolation layer, the combination of the MSE loss and KL loss, etc.

Next, I think the authors are missing quite a few references. On page 4, the authors state "In general, online time series forecasting has not been widely studied using deep learning models" but this is not true. For instance, there have been a large number of works for training recurrent neural networks in a streaming setting [3, 4]. Moreover, there are a variety of works of training deep state-space models on streaming data [5, 6].

Lastly, it is almost impossible to reproduce the results of this paper. There are no details of the architecture used in the neural ODE, the step size used in Euler integration, the number of gradient steps used during the warm-up phase, how the model is trained in the streaming setting, etc. This is even more concerning as training neural ODEs can be hairy and there exists a bevy of work on regularization approaches on stabilizing training, e.g., [1]. Moreover, the authors use an RNN for the encoder but don't mention how they handle the irregularly sampled data when it comes to the RNN encoder (for instance, the authors could be using RNNs that have been designed for irregularly sampled data, i.e., [7]). Also, I think a critical baseline is a non-continual learning model as a baseline. For instance, just training an feedforward network or RNN, as an example.


## References

[1] How to train your neural ODE: the world of Jacobian and kinetic regularization. https://arxiv.org/abs/2002.02798

[2] LFADS - Latent Factor Analysis via Dynamical Systems. https://arxiv.org/abs/1608.06315

[3] Exploring the Promise and Limits of Real-Time Recurrent Learning. https://arxiv.org/abs/2305.19044

[4] Generating Coherent Patterns of Activity from Chaotic Neural Networks. https://pmc.ncbi.nlm.nih.gov/articles/PMC2756108/pdf/nihms-137176.pdf

[5] Real-Time Variational Method for Learning Neural Trajectory and its Dynamics. https://arxiv.org/abs/2305.11278

[6] Online Variational Filtering and Parameter Learning. https://arxiv.org/abs/2110.13549

[7] Learning Long-Term Dependencies in Irregularly-Sampled Time Series. https://arxiv.org/abs/2006.04418

---

> ### Author Response · Authors · 2024-10-31
>
> We sincerely appreciate your feedback and suggestions. Your comments have helped us identify areas for clarification and enhancement in the presentation of our work. Below, we provide detailed responses to your points and outline the changes in the manuscript.
>
> **The VAE literature:**  Thank you for your valuable input regarding the notation and the clarity in our manuscript. We apologize for any confusion caused by our previous notation and have made revisions to align with the standard definition in the VAE literature. We have rewritten and summarized section 4.3, following similar related works. As Follow:
>
> >The core concept of a VAE involves two main components. First, an encoder encodes the input data $x$ into a latent variable $z$capturing its representation distribution through a probabilistic approach parametrized with $\theta$ \cite{KingmaW13}. Next, a decoder reconstructs the original input data using the latent variable $z$.
> >In more details,  the encoder $e$, which parametrised by $\theta$,  produces the outputs: the mean $Z_\mu$ and log variance $Z_\sigma$ of a normal distribution, typically assuming a prior
> $p(z)$ as a standard multivariate Gaussian distribution, denoted as  $\mathcal{N}(0,1)$.
> The following equation represents the probability function of the Gaussian distribution:
> \begin{equation}
>     q(z|x) = \mathcal{N}(Z_\mu ,Z_\sigma ),
> \end{equation}
> >here, $q(z|x)$ is the conditional distribution of the data $x$ given the latent variable $z$, and $\mathcal{N}$ is the Gaussian distribution with mean $Z_\mu$ and standard deviation $Z_\sigma$.  The latent variable $z$ is sampled from the reparameterized distribution as:
> \begin{equation}
>     Z = z_\mu  +  \epsilon \bigodot exp (0.5\cdot z_\sigma)~~,
> \end{equation}
> >where $\epsilon$ is a random sample drawn from a standard normal distribution $\mathcal{N}(0,1)$. This sampled latent variable
> $z$ is then fed into the decoder, which generates the final output  $y$ by expressing p(x|z) as:
> \begin{equation}
>       p(x|z) = \mathcal{N}({{\mu}_y}, {{\sigma}_y^2}),
> \end{equation}
>  >where $\mu_y$  and $\sigma_y^2$ are the parameters of the decoder.
>
> **Equations 3 and 5:** Thank you for pointing this out. The original equations were:
> >\begin{equation}
>     q(x|z) = \mathcal{N}(Z_\mu ,Z_\sigma ),
> \end{equation}
> \begin{equation}
>       p(x|z) = \mathcal{N}({z_\mu}', {z_\sigma}'),
> \end{equation}
>
> These equations represented the encoder and decoder of the VAE. However, we have modified them to emphasize the correct notation, as mentioned earlier, to be:
>
> >\begin{equation}
>     q(z|x) = \mathcal{N}(Z_\mu ,Z_\sigma ),
> \end{equation}
> \begin{equation}
>       p(x|z) = \mathcal{N}({{\mu}_y}, {{\sigma}_y^2}),
> \end{equation}
>
> **Reference works:**
> Thank you for your insightful comment regarding the references. We really appreciate your highlighting of these very recent works. To address this, we have expanded Section 2.2 to include the recent works.
>
>
> **Experimental details:**
> Thank you for highlighting this. The code is available on GitHub, with the link provided in the abstract. Additionally, we have added further details on the model architecture and experimental setup in Section 5.1.3.
> >**Setup and Experimental details:**
> For the encoder part of the VAE, we utilize RNN model consisting of one GRU layer with a hidden dimension of 64, followed by a linear layer that outputs a latent dimension of 64. The decoder comprises three main components: a Neural-ODE that initializes the latent state $z$, a linear layer that maps the latent state to the hidden dimension of 64, and a second linear layer that maps the hidden representation to the final output.  The dynamics of the system are integrated over time using the Euler method, with the step size determined based on the time duration from the initial time $t_0$   to the final time  $t_1$, constrained by a maximum step size of 0.05.
>
>
> >For all the experiments, we applied non-shuffled splitting to divide the data into a training set (25\% ) for initialising the model (warm-up training) and a streaming dataset (75\%) for online testing and training. We rescaled (normalised) the features using a standard normal distribution that fits on the training set only. We used a look-back window of 24 for our model and the baselines, while the learning rate was set to 0.001, and we employed the Adam optimiser for model optimisation. During the warm-up phase, we utilized a batch size of 64 and implemented early stopping after 10 epochs. To simulate a streaming environment for online learning, the batch size and epoch were set to one.
>
> **non-continual learning baseline:** Will be addressed in a separate comment.
>
> Thank you once again for your valuable feedback. We will address points 6 and 7, concerning the use of RNN baselines, in a separate official comment.

---

> > ### Author Response · Authors · 2025-01-17
> > **Non-Continual Learning baseline**
> >
> > Dear reviewer as mentioned before the following results include the performance along with the RNN baseline a non-continual learning baseline, where a basic RNN layer was applied in an environment learning setting.
> >
> > However, the performance is suboptimal due to the fact that these models are not designed for continual learning and are unsuitable for handling streaming environments effectively.
> >
> >
> > | Method      | ECL●    | ECL^    | ETTh1*   | ETTh1^   | ETTh2*   | ETTh2^   | ETTm1*  | ETTm1^   | WTH*    | WTH^    |
> > |-------------|---------|---------|----------|----------|----------|----------|---------|----------|---------|---------|
> > | RNN         | 0.576   | 33.84   | 43.36    | 1.2653   | 37.35    | 6.3711   | 41.08   | 0.470    | 0.1636  | 0.4616  |
> > | ER          | 2.8142  | 2.8359  | 1.9785   | 0.2349   | 6.7558   | 0.5044   | 3.055   | 0.082    | 0.3138  | 0.1788  |
> > | DER++       | 2.8107  | 2.81 | 1.9712   | 0.24     | 6.738    | 0.5042   | 3.0467  | 0.0808 | 0.3097  | 0.1717|
> > | FSnetNaive  | 2.9943  | 3.0533  | 2.001    | 0.2296   | 6.7749   | 0.5033   | 3.0595  | 0.1143   | 0.3843  | 0.2462  |
> > | FSnet       | 2.8048  | 3.6002  | 1.9342   | 0.2814   | 6.681    | 0.4388   | 3.0467  | 0.0866   | 0.3096  | 0.1633  |
> > | ODEStream   | 0.1173 | 4.095   | 0.0594 | 0.105 | 0.164 | 0.1879 | 0.0625 | 0.2178   | 0.0441 | 0.222    |

---

### Review · Reviewer_hmZ5 · 2025-01-06

**Summary Of Contributions:**

This work come up with a novel method by leveraging neural ODE for streaming data prediction.
- Proposed buffer free method, which is memory efficient.
- Raised several research question and showed the proposal can effectively answer these questions.

**Audience:**

Yes

**Claims And Evidence:**

Yes

**Requested Changes:**

See weakness above

**Strengths And Weaknesses:**

Strength
- Comprehensive experiment results
- Some ablation study to reveal importance of each part.

Weakness:
- One question is that what is the motivation behind the buffer free idea? In terms of memory efficiency, I can understand the motivation. However the paper also claim that it could benefit the performance, but intuitively, wouldn't it be even better to leverage some buffer, into current method? Would it be even better in terms of performance by adding a memory buffer into this design? Or because of the nature of this design, buffer is not available for it?
- Regarding the equation 13, would be better to elaborate a bit more, how was that KL divergence calculated, btw which 2 dsitribution?
- For the ablation study on the KL divergence, seems not all dataset shows significant improvement by adding KL divergence, would be good to have some insight why is this case.(e.g. on ECL, ETTH1 and ETTH2, with ITL, MSE-KL loss is on par, and on ECL, ETTH1 and ETTM1, without ITL, MSE-KL loss is on par or even worse sometimes.

---

> ### Author Response · Authors · 2025-01-17
> **Discussing the Weaknesses and Questions**
>
> We sincerely appreciate your comments and the time you dedicated to reviewing our paper. Below are our responses to your questions:
>
> * Question 1:
>
> We appreciate the reviewer’s thoughtful question regarding the motivation behind the buffer-free design. The primary motivation for eliminating the buffer is to enhance memory efficiency (as shown in Figure 8) while addressing challenges unique to streaming time series data, such as irregular sampling and concept drift (illustrated in Figures 3 and 4). The buffer-free design of ODEStream leverages neural ODEs and the temporal isolation layer to directly adapt to evolving dynamics without relying on stored historical data. This approach eliminates the need for intricate decision-making regarding buffer management and ensures the model remains responsive to real-time changes.
>
> * Question 2:
>
> The standard KL divergence of two Gaussians can be computed as:
>
> $KL = (N(x|\mu_1, \sigma_1) || N(x|\mu_2, \sigma_2))$
>
> Specifically, in our case where we leverage the VAE, the KL divergence measures the disparity between the approximate posterior distribution $q(z|x)$ and the prior distribution $p(z)$, encouraging the latent variables $z$ to follow a multivariate Gaussian prior $N(0,I)$. Here, $q(z∣x)$ represents the latent variable distribution inferred from the input data $x$ using the encoder in the VAE.
>
> So, we can get Equation 13. Two distributions are the generate latent vectors distribution $q(z∣x)$ and the unit gaussian distribution $p(z)$.
>
> * Question 3:
>
> For the ablation study presented in Table 2, for the first two rows, both the TIL layers are removed. Specifically, KL loss is not used for the first case. For the last two rows, both the TIL layers are enabled and the KL loss is disabled for the third row case. In scenarios with the TIL (the last two rows), introducing the KL loss shows on par performance on the first three dataset. One insight is the effectiveness of KL regularization can vary depending on the characteristics of the datasets and the model configurations. In datasets like ECL, ETTh1, and ETTh2, where temporal dynamics may be less complex or less prone to concept drift, the additional regularization introduced by KL divergence might not significantly enhance performance. In scenarios without the temporal isolation layer (compared the first and the second row of Table 2), with and without KL is on par on even worse is because the lack of a mechanism to focus on recent data might limit the utility of KL divergence, as the model struggles to adapt effectively to the evolving dynamics.

---

### Decision · Action_Editor_nbGM · 2025-03-19

**Recommendation:** Accept with minor revision

**Comment:**

While reviewers acknowledge the simplicity and effectiveness of the method and accompanying experiments, several areas require further attention for clarity and completeness. In particular, the writing is sometimes imprecise, and important references on streaming time series forecasting are missing. Please revise the paper accordingly, incorporating the reviewers' comments and suggestions. These refinements will strengthen the paper’s contribution to TMLR. Hence, I recommend Acceptance with the minor revision.

**Audience:**

The topic and findings are of interest to at least some of TMLR's audience.

**Claims And Evidence:**

This paper introduces ODEStream, a buffer-free continual learning method for streaming time series forecasting, leveraging neural ODEs. The proposed approach aims to improve memory efficiency and simplify deployment while handling irregular sampling in real-world data. Extensive experiments show strong performance, suggesting ODEStream is an effective solution for streaming time series tasks.

---

> ### Author Response · Authors · 2025-04-09
>
> Dear Editor,
>
> Thank you for your valuable time. The final version of the manuscript has been submitted, along with a link to the code.
>
> All reviewer comments have been addressed, as outlined in the discussion reply for each review. Several minor issues have been fixed to enhance the clarity of the article.